# Effective Mechanism To Mitigate Injuries During NFL Plays

*Abstract*—NFL(American football),which is regarded as the premier sports icon of America, has been severely accused in the recent years of being exposed to dangerous injuries that prove to be a bigger crisis as the players' lives have been increasingly at risk. Concussions, which refer to the serious brain traumas experienced during the passage of NFL play, have displayed a  dramatic rise in the recent seasons concluding in an alarming rate in 2017/18. Acknowledging the potential risk, the NFL has been trying to fight via NeuroIntel AI mechanism as well as modifying existing  game rules and risky play practices to reduce the rate of concussions. As a remedy, we are suggesting an effective mechanism to extensively analyse the potential concussion risks by adopting predictive analysis to project injury risk percentage per each play and positional impact analysis to suggest safer team formation pairs to lessen injuries to offer a comprehensive study on NFL injury analysis. The proposed data analytical approach differentiates itself from the other similar approaches that were focused only on the descriptive analysis rather than going for a bigger context with predictive modelling and formation pairs mining that would assist in modifying existing rules to tackle injury concerns. The predictive model that works with Kafka-stream processor real-time inputs and risky formation pairs identification by designing FP-Matrix, makes this far-reaching solution to analyse injury data on various grounds wherever applicable.

*Keywords*—*NFL Plays, Concussions, Defense, Offense, Formation pairs, Predictive model, FP-Matrix*

## I. INTRODUCTION

Owing to the rigorous play practices adopted in NFL games compared to contemporaries, NFL has been often touted as riskier play [4,13] with comparatively higher injury rate with the Offense players, who are generally referred to those from team with ball's possession, are highly prone to injuries than defense players who attack former to stop them scoring touchdown (similar to 'goal' in soccer). Thus, there arises the need to analyse and modify age-old rules which have been provoking the players to adopt rigorous tackles. Concussions, referring to severe brain injuries, have become an increasingly common phenomenon in NFL exhibiting a steady increase over the last six years peaking in an alarming concussion rate of 16% for the last reported 2017/18 season. This ultimately paved way for the C.T.E(Chronic Traumatic Encephalopathy) [2], a neuro-degenerative brain disease found in 96% of deceased ex-NFL players [5] and was diagnosed as the reason for death of infamous pro-NFL player, Kevin turner[6],which is generally caused by repetitive blows to the head and rigorous blows and tackles intentionally harming the health condition of the player while players have also started to report the immediate repeated concussion effects such as loss of consciousness and memory as well making a long time to recover [1].

Having acknowledged the link to C.T.E[8] from its football practice, NFL has been trying to combat the concussions via Neuro-intel's real-time tracking [3] of helmet blows to predict concussion risk rate (C.R.I) but also announced to be receptive for comprehensive analysis and suggestions to fix concussion protocol such that it would help in reducing concussions and severe injuries during plays. NFL has been amending its rules [9] and modernizing its riskier plays [10] citing concussion risks [7] continuously despite being a work in progress [11] as last reported statistics showed a surge in injuries. Being mired in handling concussions, the future of the NFL is at stake with players' safety on the line unless an extensive concussion analysis mechanism is set [12] incorporating statistical inferences from past data and different perceptions from its followers regarding head injuries as they've started to discuss widely [13].

The proposed overall research area is presented as the go-to solution that boasts of a comprehensive analytical mechanism to effectively address the aforementioned concussion crisis and related issues by incorporating pre-processed NFL punt-play datasets utilizing the available movement vectors in order to assist in predictive analytical perspective and to analyse positional variation and propose safer formation pairs by highlighting dangerous pairs via comprehensive mapping through simulation of plays.

The rest of the contents of the paper is organized as follows. In section II, the literature survey dealing with similar contemporary studies and research gap has been analysed. Section III describes the different types of experimental data that are used throughout this research. Section IV goes on to detail the phases of the methodology adopted. The findings of the research and their resultant discussions have been described in Section V of this paper. Finally, Section VI wraps up with the conclusion and future works of this research topic.

## II. RELATED WORK

### A. Injury Predictive Modelling

Analysing the injury risk encountered by a player during a particular season deriving their injury probabilities to develop a predictive model to foresee whether a player would be concussed or not in NBA [14] was presented to aid in strategic decision making throughout the season. NBA coverage, play-play data, individual workload were the sources to be collected and scraped to drive through net impact and incremental injury risk modelling utilizing the sliding window to encompass both aggregation and prediction time window to match against injury response variable. K-fold cross- validation was used to validate windows' performance whereas the Committee machine model was used to train models. Random forest was opted to counter the drawbacks of earlier drawbacks of Linear regression and SVM algorithms.

Contrary to the earlier approach, a multi-dimensional approach was opted to handle injury prediction in Soccer optimizing GPS data accumulated from training sessions and machine learning techniques [15] to predict whether a player would be concussed in the immediate game/training session he encounters. Traditional predictors were initially tested upon features extracted from the tracked data to address on its inefficiency in precision and recall. The Decision tree was chosen as the predictive algorithm. Four different datasets were constructed and the Exponential weighted moving average(EWMA) was applied to all. Recursive feature elimination with cross-validation (RFECV) was used as dimension reduction technique to realize optimal features. The model built was found to be performing accordingly on feature set. Weekly update of classifier was also found to increase the accuracy giving higher precision.

A Hierarchical multi-layered predictive system to predict injury risk among athletes considering workload assigned was proposed[17] addressing the inadequacy and inefficiency of the classic technique, ACWR as described in the above-mentioned study[15], the two-layer mechanism was proposed that encompasses injury-prone players and impacting factors and an evolving injury prediction model from that. KNN algorithm using Euclidean distance was used to build a training and testing database along with K-means classifier for benchmarking to identify overlapping features and infer them to be less impacting than the others. Output was validated with a multi-dimensional classifier. Injury prediction model implemented SVM classifier and output was normalized and validated with K-fold cross-validation culminating in a probabilistic prediction map tailored for individual player specific features. Though SVM fairly separates prediction margins, more suitable algorithms such as the Random forest could be utilized here as put forward by the aforementioned studies [14,16].

### B. Impact of positional arrangement on injuries

The Exploring movement similarity and deriving patterns prospectively could prove to be a virtue in predictive modelling. The proposed study that explores the movement similarity analysis of moving objects [19] utilizing the GIScience approach puts forward the conceptual framework to infer similarities from the movement trajectories of dynamic objects. The four-stage methodology namely framework to capture the movement of objects, defined classification to identify movement patterns, feature extraction on movements and trajectory similarity search and its evaluation were used in this workflow. Since this study explores the trajectories and mining patterns, it could be very well suited in the trajectory analysis of players in a game and understand the variation of their game formations as well that could be impactful in the occurrence of injuries subsequently.

Analysing team formation in football was made more intuitive with the approach opted i.e. Static Qualitative Trajectory Calculus(QTC) [18] which was one of Spatio-temporal methods [19] used to analyse the team formations within a game or for a season. QTC employs comparison approach by defining relative positions between players in a qualitative manner that aligns well with the way player arrange themselves on the playfield. QTC could be used in describing team formations at different time by constructing a QTC matrix at different time stamps. i.e. if the number of players in a formation is identical at each timestamp, the matrix would have the same dimensions utilizing the distances between them. Similarities among the generally defined team formations could well also be calculated and interpreted. Inferring generally accepted formations or assigning new arrangement as the formation is made possible opting for QTC based approach or building relevant matrix in the context of the higher concussions yielding formations among the rest.

## III. EXPERIMENTAL DATASET

The dataset utilized in this study was provided by NFL sourced from Kaggle platform. NFL Punt Analytics data consisted of details corresponding to games played during the 2016 and 2017 pre, regular and post seasons. This dataset encompasses a rich dataset for the aforementioned seasons describing season, games and plays on various grounds with concussion information. The data describes the player movement vectors on each play during a specific game including x coordinates, y coordinates, orientation, direction, distance covered and event.

## IV. METHODOLOGY

In this section, the methodology used for this study that includes two different modules namely predictive modelling, and formation analysis will be discussed in detail.

### A. Predictive Modelling

*1) How did we deal and explore with the enormity and the unbalanced nature of data?*

Movement vectors of each player (22 players at a play) that are retrieved from data set that had captured each player's x, y coordinates, initial direction, current direction, event specified, and the distance covered individually during that specific play along with other relevant columns such as time, position and play description, were utilized to analyse the influence of the predictors and to develop the subsequent predictive model that could distinguish the concussions among the rest. Since the data was quite large (4GB) to be processed at once, proper file format was explored and data chunking mechanism was opted to retrieve predefined chunks of data at a time to speed up process. *Feather*, which is a fast-disk format used in efficient reading and writing of data frames using defined chunk sizes, around 500000 records per chunk, was applied in this context.19 chunks were obtained, and 2 chunks were chosen to be utilized in the model building process yielding around 1000000 records. Preliminary data exploration including missing value handling and descriptive analysis were done to understand the data. Training and testing data were planned to be exported as Feather format for easy reads. On preliminary execution, it was found that the data for injury occurrence is quite low compared to the non-injuries [96.8% of 0s and 3.2% of 1s] as illustrated in Fig.1.

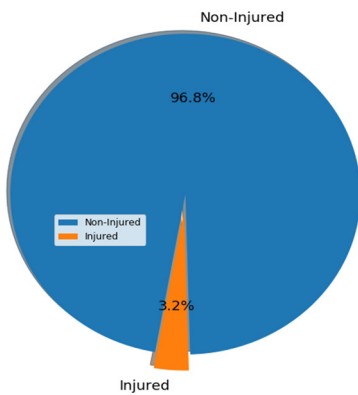

Fig.1. Distribution of Injured Vs Non-injured in data

So, the need to resample the data was realized to deal with the highly unbalanced data in order to avoid the under sampling and oversampling issues that could arise later. Despite the imbalanced nature of the data, more preferred algorithm [14], Random Forest Classifier was applied to realize the model score, 0.9681, which obviously pointed out the over fitting issue. After inspecting ROC and PXR curve, true positive rate is found to be 0. i.e. Injury detection during an injury occurrence is hardly made possible as indicated in Fig.2. (one-one detection is 0)

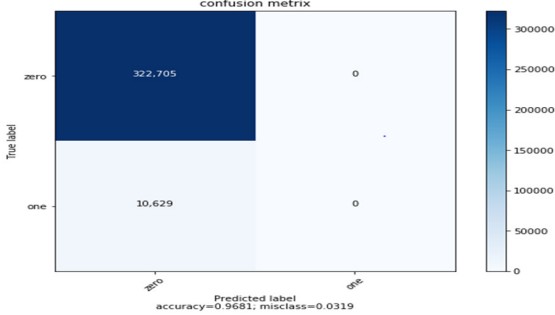

Fig.2. Confusion matrix for model on initial unbalanced data

*2) How did we apply rebalancing and distinguish the distribution of data?*

*SMOTE* (Synthetic Minority Over-Sampling Technique) from i*mblearn* was chosen as the ideal synthetic balancing technique to rebalance the data to be used later. SMOTE was applied for training data only; but not for testing set since synthetic samples created with this sampling technique aren't real but synthetic which makes them not viable for testing purposes but adequate for training. After balancing the total data (1290306 records), equal 0s and 1s were derived (645153 records each). Subsequently PCA was used to explore and illustrate the variance before and after applying SMOTE as described in Fig.3.

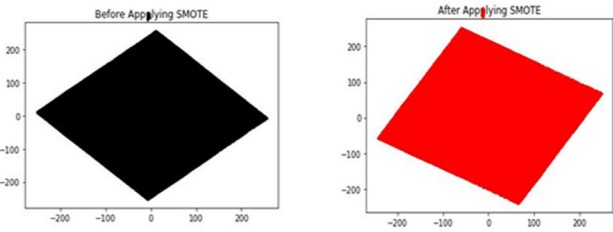

Fig.3. Before and After SMOTE variance explained using PCA

*3) Was there any decisive advantage with more features?*

Generating new columns for the model was also tried using the open source, *Feature tools*. About 25 features were generated by considering combination of 5 features namely x, y, o, dis and dir. and PCA was applied on the generated feature matrix data.

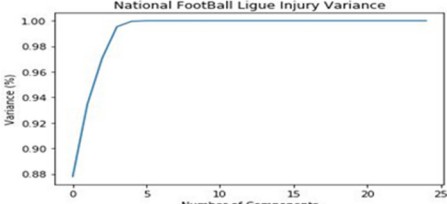

Fig.4. Variance vs Number of Components

As per Fig.4., the optimal number of components was found to be 5. i.e. by choosing 5 components, almost we could explain the variance of the injury prediction. Thereafter the feature matrix data was then split into training and testing data in 2:1 ratio and then SMOTE was applied upon it and Random Forest Classifier was applied to yield the model score (0.698,0.502) which was lesser than before suggesting the inefficiency and inadequacy of the generated features. Since the performance of the model wasn't good, the feature generation process was not successful as intended.

*4) Didn't we apply a wide range of algorithms and choose the best out of it?*

Thereafter, Random Forest algorithm is applied with hyper parameter tuning along with Grid-search with 5 folds. An accuracy of 0.639 and F1-score of 0.75 is realized from this model.

Then, K-Nearest Neighbour was tried with hyper parameter tuning.

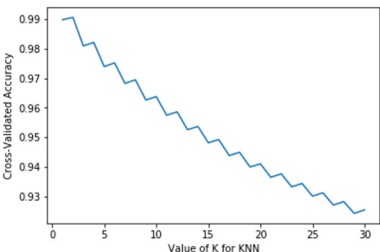

Fig.5. Value of K for KNN vs. Cross-Validated Accuracy

Even for the lowest value for K, i.e. 30, as described in Fig.5., when applied, model was found to be over fitting too much illustrating a score of 0.924.

SVM was also tried as well but that was slow on performance since the data row records are large in number and it was inferred that it didn't fit this scenario.

XG-Boost was found to be outperforming aforementioned algorithms when it was applied subsequently yielding the better accuracy of lot ,0.7959, since it follows implementation of gradient boosted decision trees that could enhance existing decision trees applicable on similar scenario [15].

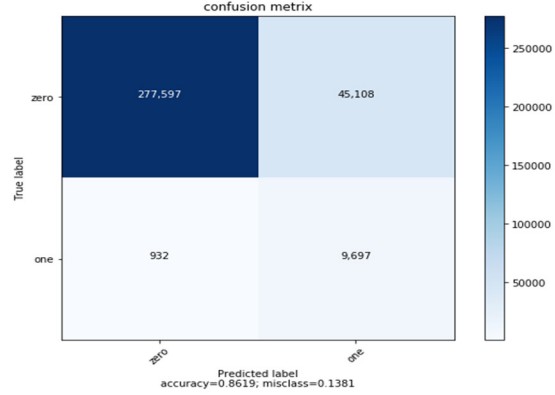

Fig.6. Confusion matrix for XGBoost applied final model

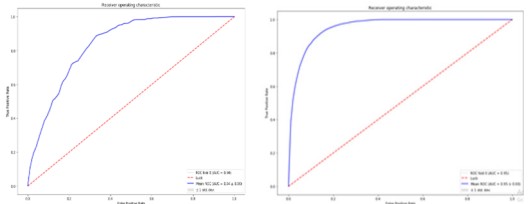

Fig.7. ROC curve compared Before and After XGBoost modelling

As illustrated in Fig.7., the ROC curve at the initial stage and the curve after XGBoost modelling explicitly differs in the smoothness of curve highlighting the better performance of the finalized model.

```
             precision    recall  f1-score   support

  NonInjured       1.00      0.86      0.92    322705
     Injured       0.18      0.91      0.30     10629

   micro avg       0.86      0.86      0.86    333334
   macro avg       0.59      0.89      0.61    333334
weighted avg       0.97      0.86      0.90    333334
```

Fig.8. Classifier report after building XGBoost modelling

After parameter tuning the above XG-Boost based model, an accuracy of 0.8618 and F1-score of 0.90 was realized and subsequently was finalized as the optimal model score to be applied for this context.

*5)    What's the novel data pipeline to put this model into practice?*

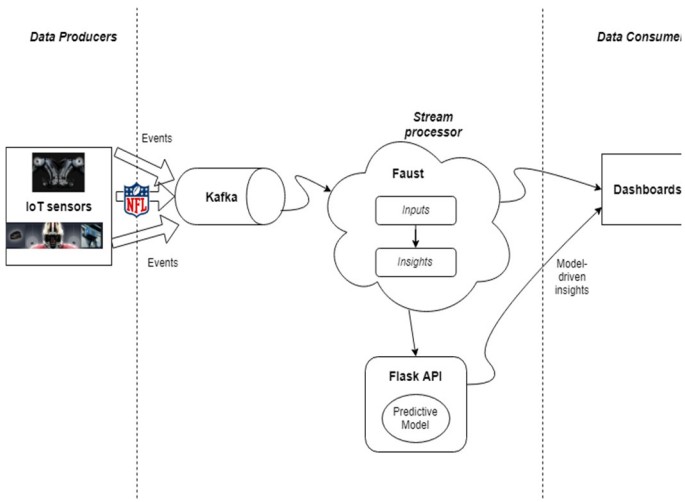

Fig.9.Proposed overall architecture for predictive modelling

Events of NFL plays that are captured from IoT sensors would be transferred to Apache Kafka (In this scenario, data provided by NFL) and subsequently they would be published and subscribed as topics. The subscribed topics would then be captured and taken to a Stream processor, Faust, to derive insights. Flask API would thereafter be utilized in getting requests from Stream processor and inbuilt deployed predictive model would be generating injury percentage corresponding to each play for every millisecond.

### B. Analyse team formations and choose optimal formations to mitigate injuries.

In this section, a novel methodology to find optimal formation pairs in NFL plays is introduced to mitigate injuries by systematically analysing the occurrence of injuries and mapping them with the formations in plays. A brief overview of formations in American football will be given first to reinstate the conceptual flow followed by the methodological steps undertaken to realize the objective of this component in detail.

*1)    Formations in American Football*
In the NFL each play starts with two main initial formations such as offensive team formation and defensive team formation. There are 16 offensive formations and 15 defensive formations. The 16 offense team formations are T formation, I formation, Single set back, Pro set, Single wing, Wildcat formation, Double wing, Short punt formation, Shotgun, Pistol, V formation, Wishbone, Flexbone, Wing T, Empty backfield, Goal-line formation, Kneel, Tackle spread. The 15 defense team formations are 4–3 defense, 6–1 defense, 3–4 defense, 2–5 defense, 4–4 defense, 5-3 defense, 6-2 defense, 38 defense (split middle), 46 defense (forty-six), 5–2 defense, Seven-man line defense, Nickel formation, Dime formation, Quarter and half-dollar formations (prevent defense), Other variants[20]

*2)    Systematic workflow*
There are three steps to be adopted to achieve the objective of obtaining optimal formation pairs. Simulating the players' movement and defining formations for each play, Building FP(Formation-Play) matrix and finally finding the optimal formation pairs to mitigate injury. The workflow then individually elaborates the methodology adopted by detailing into each subcomponent.

*a)    Simulate the players' movement and define formations for each play*
The dataset has x, y coordinates with the exact time for 22 players in each play. Each play is about 1 minute of duration. An intuitive visualization tool, Power BI, which provides functionalities to transform the Spatio-temporal data into simulated result, is utilized as the simulator in this context. The snapshot of player positions is taken from the simulated result when the play starts. Then the offensive and defensive formations are identified by comparing with predefined formations mentioned above.

### b) Build FP (Formation Play) matrix

FP matrix is a matrix of Formation Play that specifies the probability of concussed and non-concussed play against offensive and defensive pairs. The probability of occurrence range is between 0 to 1.

### c) Find optimal formation to mitigate injury

In this final step, in order to find optimal formation pairs, the probability of non-concussion should be maximum and the probability of concussion has to be minimal. The significance of formation pairs is measured by how often the formation pairs are used. Hence, optimal formation pairs as well as dangerous set of pairs can be identified adopting the proposed flow of steps.

## V. RESULTS AND DISCUSSION

Since predictive modelling is one of the primary components of the proposed research, the results of the predictive modelling are well-assessed. On application of different algorithms such as Random-forest, SVM and KNN, different problems were encountered. SVM was awful in performance requiring lots of time whereas Random forest displayed poor scores. KNN, despite depicting better accuracy score of the lot, wasn't chosen to be fitting the narrative illustrating the impending concern for overfitting issues. XGBoost algorithm was chosen as the appropriate model to be applied adequately displaying higher scores without overfitting concerns.

|  | Random-Forest | K-Nearest Neighbor | Xg-Boost |
|---|---|---|---|
| f1-score | 0.75 | 0.94 | 0.90 |
| precision | 0.96 | 0.98 | 0.97 |
| recall | 0.64 | 0.92 | 0.86 |
| Accuracy-Score | 0.639 | 0.924 | 0.861 |

Fig.10. Different Algorithms applied with respective model scores.

Since injury predictive mechanism relies on the sensitivity (the proportion of detecting the injured/concussed players correctly among those who're injured actually), due care should be taken to maximize the sensitivity along with model score by increasing the volume of historical data and training data on the Big Data Machine Learning platform. The proposed injury predictive model works as an injury predictor with fixed time stamp identifying the concussions occurred time by providing an injury occurring percentage to each play. Defining a threshold to the aforementioned injury percentage outputs by model, we could distinguish the awful concussions from the rest of the injuries subsequently proving beneficial in providing pre-attention and prevention to players succumbing to dangerous injuries making the game safer.

Kafka is proposed as the message queue for streaming data and the player movement vectors will be captured accordingly. Through Faust, a Stream processor library to port ideas from Kafka to python, the movements of 22 players in a play is input to the model. Utilizing the batch window mechanism, fixed, non-overlapping time intervals, Flask, micro web framework, is used to acquire requests and deliver predictions on each millisecond timestamp. This proposed architecture could very well be applied in any injury predictive mechanism realizing the conceptual framework behind its application.

In the formation analysis module, the following set of results were derived and the need to handle and tweak the rules corresponding to these formation pairs were realized. The top 10 optimal formation pairs are illustrated in Fig.11 and the top 10 risky formation pairs are illustrated in Fig.12.

| index | def_formation | off_formation | FC0 | FC1 | TotFreq | PC0 | PC1 |
|---|---|---|---|---|---|---|---|
| 39 | 5-3 Defense | Single Set Back | 8 | 10 | 18 | 0.080808 | 0.322581 |
| 14 | 4-3 Defense | I Formation | 5 | 0 | 5 | 0.050505 | 0 |
| 32 | 5-3 Defense | I Formation | 5 | 2 | 7 | 0.050505 | 0.064516 |
| 24 | 46 Defense | Single Set Back | 4 | 0 | 4 | 0.040404 | 0 |
| 37 | 5-3 Defense | Shotgun | 4 | 0 | 4 | 0.040404 | 0 |
| 16 | 4-3 Defense | Single Set Back | 4 | 1 | 5 | 0.040404 | 0.032258 |
| 8 | 3-4 Defense | Single Set Back | 3 | 0 | 3 | 0.030303 | 0 |
| 35 | 5-3 Defense | Proset | 3 | 0 | 3 | 0.030303 | 0 |
| 56 | 6-2 Defense | T Formation | 3 | 0 | 3 | 0.030303 | 0 |
| 22 | 46 Defense | I Formation | 2 | 0 | 2 | 0.020202 | 0 |

Fig.11. Top 10 optimal formation pairs derived

| index | def_formation | off_formation | FC0 | FC1 | TotFreq | PC0 | PC1 |
|---|---|---|---|---|---|---|---|
| 27 | 5-2 Defense | Pistol | 0 | 3 | 3 | 0 | 0.096774 |
| 45 | 6-1 Defense | I Formation | 0 | 3 | 3 | 0 | 0.096774 |
| 19 | 4-4 Defense | Wing T | 0 | 2 | 2 | 0 | 0.064516 |
| 21 |  | 46 Single Set Back | 0 | 1 | 1 | 0 | 0.032258 |
| 47 | 6-1 Defense | Short Punt | 0 | 1 | 1 | 0 | 0.032258 |
| 28 | 5-2 Defense | Single Set Back | 1 | 2 | 3 | 0.010101 | 0.064516 |
| 36 | 5-3 Defense | Shot Punt | 1 | 2 | 3 | 0.010101 | 0.064516 |
| 40 | 5-3 Defense | Single Wing | 1 | 2 | 3 | 0.010101 | 0.064516 |
| 63 | Seven-man line D | Pistol | 1 | 0 | 1 | 0.010101 | 0 |
| 64 | Seven-man line D | Wing T | 1 | 0 | 1 | 0.010101 | 0 |

Fig.12. Top 10 risky formation pairs derived

## VI. CONCLUSION AND FUTURE WORKS

Throughout this research study, it is proven that the necessity to build a comprehensive injury predictive mechanism to mitigate injuries dealing with multiple sources and different components that could redefine the way that the injury control mechanism to be performed in the near future. The proposed study highlights the benefits to be yielded having roots implanted in different components namely exploring and pre-processing data, predictive modelling, formation analysis using simulation & FP-matrix. Injury predictive modelling details into extensive architecture to be followed in predicting injury percentage from streaming feeds where as dangerous formation pairs yielding higher concussion rate are mined and chosen through simulation and mapped then to FP matrix highlighting the need for authorities to relook into it and regulate. Experimental results have validated the theoretical framework specified. This proposed workflow could be utilized for any games in developing their injury predictive mechanism strengthening their existing injury protocol by only tweaking little bit thus proving to be an effective solution wherever applicable. Just in the case of Srilankan context, National Rugby team is unable to cope with the withdrawal of players due to injury fears in the absence of proper injury control mechanism and protocols[21,22].Thus, our proposed comprehensive solution could very well be applied in similar contexts and

could be developed for different concussing games accordingly.

Other than this, on the technical context, the future works could shed light upon finding proper data capturing mechanism to regulate inputs by introducing IoT concepts into existing mechanism and enhance the modelling approach by trying out multiple classification algorithms. Also, due care should be put in using optimization algorithms like genetic algorithms to derive optimal formation pairs and they might yield better results than current context. Automating the offensive and defensive formation classification to reduce the time and cost would be another future work that would significantly enhance the value of the solution.

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
