# OpenReview forum: "Effective Mechanism to Mitigate Injuries During NFL Plays "
_ICLR.cc/2020/Conference — Reject_

### Official Review · AnonReviewer1 · 2019-10-23
**Official Blind Review #1**

**Rating:** 1

**Review:**

The paper aims to improve injury prediction and modeling in the National Football League (NFL) using machine learning. Unfortunately, the lack of clarity in the paper and poor writing prevents me from writing a thorough review. Given that the authors seem to apply off-the-shelf machine learning algorithms to NFL data, and spend a significant amount of time discussing low-level details of data storage, preprocessing, and underlying hardware/software pipelines, this paper does not appear to be a good fit for ICLR and would be better served at either a data science conference, or alternatively a sports medicine/science conference.

**Experience Assessment:**

I do not know much about this area.

**Review Assessment: Checking Correctness Of Derivations And Theory:**

N/A

**Review Assessment: Checking Correctness Of Experiments:**

N/A

**Review Assessment: Thoroughness In Paper Reading:**

I made a quick assessment of this paper.

---

### Official Review · AnonReviewer4 · 2019-10-25
**Official Blind Review #4**

**Rating:** 1

**Review:**

This paper introduces a machine learning pipeline for injury prediction in NFL events. The paper discusses several system settings on data streaming and processing, along with model selection and other hyper-parameter tuning details. The problem itself is very important. However, there are several disadvantages of the current status of the paper.

The writing of this paper needs to be largely improved. The description is very redundant and the texts are very hard to read. Therefore, it makes the paper much less clear to the reader.

Second, the proposed system is not related to the focus of ICLR and it lacks the novelty. Both the data processing and model selection methods are already well-known practicies. Experiments are also not established well enough to demonstrate the advantage of the proposed solution to this problem.

Due to the above issues, I think the paper is not ready for publication.

**Experience Assessment:**

I have published one or two papers in this area.

**Review Assessment: Checking Correctness Of Derivations And Theory:**

I carefully checked the derivations and theory.

**Review Assessment: Checking Correctness Of Experiments:**

I carefully checked the experiments.

**Review Assessment: Thoroughness In Paper Reading:**

I read the paper thoroughly.

---

### Official Review · AnonReviewer2 · 2019-10-26
**Official Blind Review #2**

**Rating:** 1

**Review:**

The study analyzes NFL data to mitigate injuries of NFL players. Methods such as K-NN, XGBoost, and SVM are used for the analysis and predictive modeling.
Although the research is important for NFL and sports industries, there are several problems as an ICLR paper:
First, it is not formatted correctly. I guess using single column, the content may be overlength. Second, the methods used in the paper are conventional and there is no novelty from the algorithmic perspective. Third, the findings is useful within the sports industry, but does not conveys much insight for the ICLR community.


**Experience Assessment:**

I do not know much about this area.

**Review Assessment: Checking Correctness Of Derivations And Theory:**

I assessed the sensibility of the derivations and theory.

**Review Assessment: Checking Correctness Of Experiments:**

I assessed the sensibility of the experiments.

**Review Assessment: Thoroughness In Paper Reading:**

I read the paper at least twice and used my best judgement in assessing the paper.

---

### Decision · Program_Chairs · 2019-12-19

**Decision:**

Reject

**Comment:**

All reviewers recommend reject, and there is no rebuttal.